# A Narrative Review of the Efficacy of Interventions for Emotional Dysregulation, and Underlying Bio–Psycho–Social Factors

**DOI:** 10.3390/brainsci14050453

**Published:** 2024-04-30

**Authors:** Thomas Easdale-Cheele, Valeria Parlatini, Samuele Cortese, Alessio Bellato

**Affiliations:** 1School of Psychology, University of Southampton, Southampton SO17 1BJ, UK; tec1g22@soton.ac.uk (T.E.-C.); v.parlatini@soton.ac.uk (V.P.); samuele.cortese@soton.ac.uk (S.C.); 2Centre for Innovation in Mental Health, University of Southampton, Southampton SO17 1BJ, UK; 3Department of Child and Adolescent Psychiatry, Solent NHS Trust, Southampton SO19 8BR, UK; 4Department of Clinical and Experimental Sciences (CNS and Psychiatry), Faculty of Medicine, University of Southampton, Southampton SO17 1BJ, UK; 5Department of Child and Adolescent Psychiatry, Hassenfeld Children’s Hospital at NYU Langone, New York University Child Study Center, New York, NY 11042, USA; 6DiMePRe-J-Department of Precision and Regenerative Medicine-Jonic Area, University of Bari “Aldo Moro”, 70100 Bari, Italy; 7Institute for Life Sciences, University of Southampton, Southampton SO17 1BJ, UK; 8School of Psychology, University of Nottingham Malaysia, Semenyih 43500, Malaysia; 9Mind and Neurodevelopment (MiND) Interdisciplinary Cluster, University of Nottingham Malaysia, Semenyih 43500, Malaysia

**Keywords:** emotion, regulation, dysregulation, intervention, pharmacological, non-pharmacological, psychosocial

## Abstract

In this narrative, comprehensive, and updated review of the literature, we summarize evidence about the effectiveness of interventions aimed at reducing emotion dysregulation and improving emotion regulation in children, adolescents, and adults. After introducing emotion dysregulation and emotion regulation from a theoretical standpoint, we discuss the factors commonly associated with emotion regulation, including neurobiological and neuropsychological mechanisms, and the role of childhood adverse experiences and psycho–social factors in the onset of emotion dysregulation. We then present evidence about pharmacological and non-pharmacological interventions aiming at improving emotion dysregulation and promoting emotion regulation across the lifespan. Although our review was not intended as a traditional systematic review, and the search was only restricted to systematic reviews and meta-analyses, we highlighted important implications and provided recommendations for clinical practice and future research in this field.

## 1. Introduction

Emotion regulation (ER) can be conceptualized as the set of strategies to recognize, attend, appraise, and respond to emotional experiences [1,2]. ER is considered a central tenant of emotional competence [3], which entails the use of contextually appropriate emotional knowledge and coping strategies, especially in response to stressful events and negative emotions [3,4]. A categorization of ER strategies as *adaptive* or *maladaptive* is often portrayed in the literature [5]. For example, cognitive reappraisal (which involves changing the cognitive meaning of a situation or event and its emotional valence), acceptance, and attempting to change the nature of a negative situation by eliminating the source of stress (problem solving) are associated with better interpersonal functioning, greater wellbeing, and positive affect. Conversely, emotional avoidance (ignoring the source of stress) and suppression of facial expressions and emotional behaviors have been found to lead to increased depression and anxiety, negative mood, lower social functioning, and reduced psychological well-being [6]. Nevertheless, regulatory flexibility is proposed as a more comprehensive framework through which the adaptability of ER strategies shall be considered, accounting for individual differences and the situation surrounding the emotional experience [7,8]. This framework specifically highlights the importance of one’s sensitivity to contextual and temporal demands, the repertoire of ER strategies available to the individual, and the ability to select an appropriate strategy and monitor feedback following its deployment [7]. Flexibly implementing a wide repertoire of ER strategies allows one to control behavioral, cognitive, and physiological elements in the interest of managing engagement or disengagement, appraisal, and response to emotional content to regulate the emotional experience [9], enabling adaptive and goal-directed behavioral responses to emotional stimuli and information [10].

Conversely, emotional dysregulation (ED) is associated with heightened emotional sensitivity [11], irritability (namely, increased proneness to experience and/or express negative emotional states [12]), and reduced ability to recognize emotional states, although it cannot be simply conceptualized as the opposite of ER [4,11]. ED leads to dysfunctional and disproportionate responses to emotionally salient stimuli [11,13,14,15] by interfering with cognitive processes in appraising and attending to emotional content, behavioral responses, and physiological regulation [11,15,16]. It is often associated with substance misuse, social avoidance, self-injurious thoughts and behaviors (some of which implemented as maladaptive ER strategies), or challenges with engaging in goal-directed or task-driven behavior [11,15,17,18]. ED is considered a secondary symptomatologic component of different mental disorders, including Attention-Deficit/Hyperactivity Disorder (ADHD), Autism Spectrum Disorder (ASD), Major Depressive Disorder (MDD), Borderline Personality Disorder (BPD), schizophrenia, and Bipolar Disorder (BD) [15,19,20,21]. Importantly, ED has been proposed to be a transdiagnostic (namely, present across different disorders) contributor to the onset and maintenance of internalizing and externalizing symptoms [22].

## 2. Search Strategy

Building on evidence from recently conducted systematic reviews and meta-analyses (primarily summarizing randomized controlled trials and cohort interventional studies), we present a narrative review of interventions that are available and used clinically to decrease behavioral symptoms of emotion dysregulation, in children/adolescents and adults. We also focus on interventions aimed at promoting emotion regulation. We identified the studies via a systematic screening of records published in PubMed up to 24th April 2024, via the following search strategy: “(emotion* regulation OR emotion* dysregulation) AND (treatment* OR intervention* OR management OR therapy)”. The retrieved articles were screened by T.E.-C. and cross-checked by A.B., to identify available research on factors underlying ER and ED (reported in Section 3) and evidence of the effectiveness of different pharmacological (Section 4) and non-pharmacological interventions (Section 5) for reducing ED or promoting ER.

## 3. Factors Associated with Emotion Regulation and Dysregulation

To better understand the nature and the mechanisms underlying interventions for ED, it is important to first analyze what factors are associated with ER and ED. These include childhood adverse experiences (e.g., trauma, maltreatment, institutionalization, illness), and related neurobiological, neurodevelopmental, and psycho–social factors.

### 3.1. Neurobiological, Neurocognitive, and Neurophysiological Factors

Genetic and environmental factors (including childhood adverse experiences [23], as discussed below, and illness) may interplay and alter trajectories of brain development and/or functioning, influencing the acquisition of ER skills across development [24]. The main neural systems implicated in ER are the dorsal and ventral regions of the anterior cingulate cortex, the dorsolateral and ventrolateral/ventromedial regions of the prefrontal cortex (PFC), the pre-supplementary and supplementary motor areas, together with the insula, the amygdala, and the periaqueductal grey (see Figure 2 published in [25], for a graphical overview). The optimal development of the PFC (which is not fully mature until young adulthood) is particularly relevant as it supports several processes linked to ER, such as effortful control [26], executive functions [27], and goal-directed behavior [28]. Moreover, as structural and functional connectivity between pre-frontal, limbic, and brainstem systems gradually increase throughout development, basic extrinsic ER strategies (such as calming down, after experiencing stress, with a pacifier or when hugged by parents) evolve into more elaborated and intrinsic strategies at a later age [29]. Difficulties in ER have been associated with decreased connectivity between prefrontal areas and the limbic system, particularly the amygdala [27,30]. Conversely, effective ER depends upon developmentally mature connectivity between these cortical and subcortical systems [31]. While prefrontal systems are more involved in the control of the emotional experience, the amygdala is involved in the generation of that experience and emotional expression [32]. However, it is worth noting that the connectivity between the PFC and the amygdala is bidirectional. The amygdala also transmits information on the emotional valence of sensory information to the frontal cortex, which then supports the selection of the most appropriate response and/or behavior (e.g., implementing a certain ER strategy). Alterations in the development and/or functioning of prefrontal neural systems are commonly observed in mood disorders [28] and in externalizing disorders such as ADHD [30], in which ED plays an important role in the onset and maintenance of symptoms. Associations between ED and chronic inflammation have also been found [33], suggesting that chronic inflammation since early life could be a key factor impacting the development of brain structures involved in ER and, therefore, contributing to the onset and maintenance of ED and internalizing/externalizing symptoms.

Another neurobiological system possibly involved in ER and ED is the hypothalamic–pituitary–adrenal (HPA) axis, a group of endocrine structures which—together with the hypothalamic–pituitary–gonadal (HPG) reproductive axis [34]—instigates physiological responses to stressors. The HPA axis is responsible for initiating the release of a cascade of hormones, eventually culminating in the release of cortisol into the bloodstream, in response to threatening stimuli [26,35,36]. Braquehais and colleagues [37] conceptualized that responses to stressors occur in two stages. The first stage is immediate and acute, instigated by the sympathetic–adrenal–medullary axis, while the slower second stage involves the HPA axis (and the release of cortisol), contributing toward functional changes in the central and peripheral nervous systems. The PFC and the limbic system (including the amygdala) modulate both phases of the stress response. Another critical component is the serotonergic system, which is interlinked with the HPA axis [38]. In moderately stressing conditions, serotonin has an inhibitory function on the frontal cortex [35,37], promoting calmness, reduced anxiety and dysphoria, and stable/adaptive allocation of attentional resources for information processing and implementation of behavioral and/or emotional responses. In severely stressful conditions, like during adverse or traumatic experiences, serotonin is released over multiple brain regions, to compensate for elevated levels of stress. However, when the experience of stress is persistent or chronic, excessively elevated serotonin levels eventually leads to de-sensitization of serotonin receptors, leading to further reduced serotonin release and consequent impaired ability to reduce cortisol in the bloodstream. Overactivity of the HPA axis has been found associated with impaired self-regulation [26] and self-control [35], and ED [35,36,37]. Consistently, positive correlations have been found between increased cortisol and behavioral markers of ED [35].

The autonomic nervous system (ANS) is also likely to play an important role in ER and ED, since it is responsible for the modulation of physiological processes, including blood pressure and heart rate (HR), respiration, sweating, and thermoregulation. The ANS is comprised of different branches, including the sympathetic, the parasympathetic, and the enteric nervous systems (see [29] for a more detailed overview). When experiencing stress and during situations that require fast allocation and mobilization of energetic resources, the activation of the sympathetic branch of the ANS produces excitatory effects on bodily systems, such as increases in HR, galvanic skin responses, faster respiration rate, and pupil dilations. Conversely, the parasympathetic branch is mainly responsible for cardiac control through the *vagus* nerve, predominantly via slow breathing [39,40,41,42]. In situations perceived as unsafe or dangerous, increased sympathetic activation and reduced parasympathetic regulation (reflected in reduced activation of the prefrontal cortex and increased activation of the limbic and other subcortical systems involved in emotion generation) trigger a stress response, causing anxiety, distractibility, and impulsive behaviors (fight or flight response) or, vice versa, immobilization and social withdrawal (freeze response). While these are normal stress responses when facing dangers and threats, chronic exaggerated sympathetic reactivity or blunted parasympathetic activation may lead to recurrent anxiety and stress, and difficulties to adaptively regulate emotions. It has been suggested that better ability to self-regulate in challenging everyday life situations is reflected in increased heart rate variability (HRV) [42,43]. HRV is the variation in the time interval between heartbeats [44], and it is thought to reflect parasympathetic regulation of the heart via the *vagus* nerve [45]. Reduced HRV has been found associated with poorer emotional functioning and reduced self-regulation, thus it has been proposed to be a transdiagnostic marker of psychopathology and reduced social functioning [46]. Importantly, reduced HRV (especially at rest) has been associated with increased ED [19] and increased self-harm and suicidal ideation [47], in children and young people.

### 3.2. Childhood Adverse Psycho–Social Experiences and Factors

The maturation and acquisition of ER skills in children is primarily enabled by child-caregiver relationships characterized by secure attachment [48,49]. Parents and caregivers support the child when emotionally activated, typically through a process of co-regulation [49]. According to social learning theories, caregivers intimately model and scaffold ER processes, so that the child can learn to recognize, understand, and adaptively respond to the emotional content [48]. Thus, the caregiver’s own capacity to process emotional content, and to use adaptive ER strategies to cope with negative emotions, is critical to facilitate co-regulation and promote the development of ER skills in children [48,49].

Experiences of childhood maltreatment, emotional or physical abuse, or trauma exposure, have consistently been found correlated with ED in children, adolescents, and adults [24,48]. Within child–caregiver relationships characterized by maltreatment and insecure attachment, the acquisition of adaptive ER strategies is disrupted psychologically, due to the child lacking the opportunities and safety to learn adaptive ER strategies from their caregiver [24]. Additionally, childhood maltreatment is often associated with parental ED [50]. Furthermore, experiencing repeated trauma, known as poly-victimization, in youth, is associated with a greater likelihood of developing ED or greater severity of ED [24]. Importantly, there is also evidence that discrimination and exclusion (e.g., due to race or ethnicity) can exacerbate ED in children and young people [51]. Children who are forced to migrate or relocate, usually due to violence, war, and/or poverty, are at higher risk of psycho–physiological dysregulation and ED [52], due to the adverse and developmentally disruptive experiences prior to and during fleeing [53]. The risk of dysregulation is exacerbated if these children are separated from their caregivers, which most often occurs at international borders [52].

## 4. Pharmacological Interventions

Medication is often used as a treatment for mental disorders. As illustrated in more details below, there is evidence that medications commonly used to treat internalizing and externalizing symptoms (e.g., depression, ADHD, and psychotic symptoms) have beneficial effects for reducing ED and promoting ER. Pharmacological options that can have a beneficial effect on ED include antidepressants, antipsychotics, and ADHD medication.

### 4.1. Adults

Antidepressant medications are widely used to treat depression and other mood-related disorders [54]. We identified two meta-analyses investigating the effects of antidepressants on ER and ED [54,55]. Outhred and colleagues [55] presented a meta-analysis consisting of 11 fMRI studies investigating the effects of single doses of selective serotonin reuptake inhibitors (SSRIs) or noradrenergic reuptake inhibitors (NRIs) on emotion-processing tasks. They found that SSRIs decreased activation of the amygdaloid–hippocampal region, and this may be accompanied by decreased early attentional reactivity to emotional stimuli. Similarly, NRIs were found to reduce emotional reactivity by increasing activation of the cingulate cortex and thalamus. The authors conceptualized that NRI modulation of these medial regions is likely to reflect reduced attention towards emotional stimuli, which facilitates decreased reactivity and subsequently enhanced ER. Despite the differential mechanisms for improving emotional reactivity, SSRIs were shown to be superior to NRIs in terms of adaptively adjusting region-specific activation across the cortex, which the authors suggest may be a factor contributing toward the superiority of SSRIs in terms of symptomatic regulation/outcomes [55]. Furthermore, an interaction effect between the antidepressant administered and the task type was found. This interaction suggested that the effects on the amygdaloid–hippocampal region and cortical regions, including the frontal cortex, of SSRIs are more independent from the stimulus type. Effects of SSRIs were demonstrated in response to emotional pictures and faces, whereas those of NRIs were limited to emotional pictures only. Similarly, Ma [54] conducted a meta-analysis consisting of 60 fMRI studies (1569 participants with and without mood disorders) to investigate the influence of long-term use of second-generation antidepressants, SSRIs, and serotonin–norepinephrine reuptake inhibitors (SNRIs), on emotional processes. It was found that SSRIs and SNRIs augmented brain activity in response to positive emotions and decreased activity in response to negative emotions, within a circuitry encompassing the amygdala, insula, and ACC. Unfortunately, these meta-analyses did not focus on investigating the effects of SSRIs or NRIs on other markers of ED or ER (e.g., behavioral or physiological).

A large randomized controlled trial (RCT) [56] examined the implementation of adaptive and maladaptive ER strategies (e.g., cognitive reappraisal and expressive suppression, respectively) before and after 8-week treatment with SSRIs and NRIs (escitalopram, sertraline, or venlafaxine extended release) in 1008 adults with MDD. Significant decreases in the use of expressive suppression and significant increases in the use of cognitive reappraisal were found following treatment with antidepressants, suggesting that treatment significantly improved ER skills. This was highlighted by the finding of a secondary regression analysis showing that changes in ER significantly predicted improvement of depressive symptoms at Week 8 of antidepressant treatment [56]. However, there was no control arm in this RCT (since medications were compared to each other), which is an important limitation to consider when interpreting these findings.

Bergamelli and colleagues [57] reviewed 11 studies (not only RCTs, but also longitudinal and cross-sectional studies) to evaluate the impact of lithium on brain functional activity and connectivity (but not on behavioral/clinical measures of ED/ER). They reported that lithium treatment reduced functional alterations, i.e., upregulated functional connectivity between the amygdala and the PFC, and increased activation in brain regions involved in ER, including the PFC, cingulate cortex, and amygdala, during emotionally activating tasks, in adults with BPD. The authors interpreted these findings as indicating that lithium enhances connectivity and modulation of prefrontal regions and interconnected and subcortical areas implicated in the processing of emotionally activating stimuli, to facilitate the implementation of effective ER strategies.

Medication effects on ED have also been observed in people with ADHD. Lenzi and colleagues [58] conducted a meta-analysis to investigate the efficacy of stimulant and non-stimulant medications on ED in adults with ADHD. Overall, 21 double-blind RCTs of methylphenidate and lisdexamfetamine (stimulants), and atomoxetine (non-stimulant selective norepinephrine reuptake inhibitor) were included. All three medications were superior to placebo in reducing ED severity for investigator-rated or self-reported symptoms in adults with ADHD. However, effect sizes were medium for lisdexamfetamine (d = 0.5), small-to-medium for methylphenidate (d = 0.3), and small for atomoxetine (d = 0.24). Lenzi and colleagues highlighted the need for head-to-head trials and a network meta-analysis to draw meaningful conclusions regarding medication ranks [58].

### 4.2. Children and Adolescents

In relation to pharmacological interventions for children and young people, some studies investigated the effects of antipsychotics in ameliorating ED. In a recent review on the management of ED in pediatric populations, Sorter and colleagues [15] highlighted that one double-blind study found that haloperidol, a first-generation antipsychotic, is effective in treating conduct disorder (CD) and aggression, which are both characterized by ED, in children. However, they warned against adverse effects, including sedation, drooling, and dystonic reactions. Another first-generation antipsychotic, molindone, has been shown to be effective In treating impulsive aggression when administered in an extended-release preparation. Moreover, its efficacy and discontinuation rates were similar to those of olanzapine and risperidone, second-generation antipsychotics. Clozapine and risperidone have also been shown to reduce aggression in children with CD and improve (especially, clozapine) social, occupational, and psychological functioning. Lastly, risperidone was found effective in reducing irritability and frequency of temper-based outbursts in children with severe mood dysregulation.

Within ADHD medications, methylphenidate has been shown to improve ED, impulsivity, and aggression, in children and young people with ADHD, and to reduce aggressive symptoms in children with CD, among whom comorbidity with ADHD is common [15]. Furthermore, Salazar de Pablo and colleagues [59] recently conducted a meta-analysis to evaluate the efficacy and predictors of response for a broad range of pharmacological interventions to treat ED in autistic children and young people. This mainly included antipsychotics, neuropeptides, ADHD medications, fatty acids, glutamatergic blockers, opioid antagonists, and diuretics. The meta-analysis consisted of 45 placebo-controlled RCTs (2856 participants), most of which were conducted in children/adolescents. Compared to placebo, pharmacological interventions overall significantly improved ED in those with ASD (d = 0.61). Among antipsychotics, aripiprazole and risperidone—individually—were superior when compared to placebo. However, the authors warned about tolerability issues, with weight gain and restlessness being common side effects. Of note, higher baseline ED severity was associated with greater response to pharmacological interventions for ED in this population, and there was not a statistically significant moderating effect of age.

## 5. Non-Pharmacological Interventions

Recent evidence synthesis studies (see, for example, [60]) have shown the benefits of implementing non-pharmacological interventions, both in combination with medication and as standalone treatment, in people experiencing ED. As summarized in more details below, the most used and effective non-pharmacological interventions for ED (especially, for adults) are cognitive behavioral therapy (CBT), dialectical behavioral therapy (DBT), and mindfulness-based therapy (MBT). There is also evidence of benefits associated with the adaptation of these traditional protocols (see the Unified Protocol for Emotional Disorder), the delivery of interventions in group settings (which promote interpersonal validation), the integration of biofeedback components (e.g., HRV biofeedback), and the delivery of the interventions via digital technologies (e.g., mobile or virtual reality).

### 5.1. Adults

#### 5.1.1. Cognitive Behavioral Therapy

Cognitive behavioral therapy (CBT) is founded on the premise that psychopathology and psychological distress arise from and are maintained by maladaptive cognitions relating to general beliefs or schemas. These give rise to automatic cognitions in specific situations that may lead to maladaptive behavioral responses [61]. Thus, as an intervention, CBT targets maladaptive cognitions and schemas through collaborative problem-solving processes and strategies that challenge and test the validity of the client’s thoughts and beliefs. Therefore, the maladaptive behavioral patterns relating to specific situations are modified [61]. Sloan and colleagues conducted a systematic review of interventions for ED, which addressed difficulties including rumination, behavioral and experiential avoidance, and suppression [62]. Most of the 67 identified studies lacked methodological quality since; of these, 29 utilized open trial designs. Moreover, of the remaining 38 studies that utilized a control condition, only 14 compared an active treatment to another empirically established treatment, and the remaining 24 studies utilized an active control, TAU, or waitlist comparison. Nonetheless, the authors concluded that CBT was effective in significantly reducing overall deficits in ER (Cohen’s d = 1.25, large effect), improving ED—although with smaller effect (d = 0.28)—alongside improving depressive symptoms and general psychopathology. In relation to the mechanisms underlying such improvements, CBT was reported to improve specific ER skills, including emotional awareness, clarity, and understanding. In line with these findings, studies that paired CBT with integrative training of emotional competencies (ITEC) and ER skills produced significantly larger improvements in ER (d = 2.34), depression (d = 2.67), and general psychopathology (d = 2.58) compared to CBT alone [62]. The stronger effects of CBT+ITEC programs could relate to the integration of intensive learning, and practicing of general ER skills, and the use of various techniques from CBT, dialectical behavioral therapy (DBT), mindfulness-based intervention (MBI), empathy training, emotion-focused therapy, and problem-solving therapies [63]. Specifically, psychoeducation helps to explain the origins, functions, and mechanisms underlying emotional reactions. Vicious negative cycles, which are the key in the long-term maintenance of negative emotions, are presented. Lastly, techniques and strategies to interrupt these negative cycles are taught, and clients are encouraged to regularly practice and apply them when they experience difficulties with regulating their emotions.

#### 5.1.2. Dialectical Behavioral Therapy

Dialectical behavioral therapy (DBT) is a transdiagnostic intervention for those clinical presentations regarded as challenging to treat [64]. ED is a prominent feature for most of the disorders in which DBT has been established as an effective treatment, such as BPD [64,65]. According to the DBT bio–social theory, BPD is primarily characterized by pervasive ED, which results from a biological predisposition to a high degree of emotional vulnerability and invalidating environments [66]. Hence, ED is often recognized as the target of DBT; therefore, ER could be *one of*—or perhaps *the*—mechanism underlying patient improvements in DBT [64,65].

DBT conceptualizes that emotions go beyond a phenomenological experience, but rather encompass a full system [65]. The full system of emotion, initiated by emotionally salient stimuli, is comprised of neuro-chemical and physiological changes, and emotion-related action tendencies, which refer to the preparedness to establish, maintain, alter, or sever one’s relationship with the environment to contribute toward enhancing survival chances [65]. Hence, emotions are informative and instructive, informing individuals about their needs and how to appraise and respond, as well as affirming, or not, interpretations and perceptions of events. Yet, crucially, the information produced by an individual with ED can result in dysfunctional behavioral responses to control the emotional experience.

DBT is theorized to promote learning and integration of effective responses to be implemented during emotional experiences, adjunctively promoting a shift away from ineffective or maladaptive responses [65,67]. Conceptually, effective responses in DBT result from the blending of acceptance-based approaches, inspired by Zen and other contemplative practices, with cognitive–behavioral approaches. The dialectical philosophy underpinning DBT stresses the importance of balancing acknowledgement, validation, acceptance and, conversely, change. Practically speaking, DBT is helpful for improving ER because it encourages clients to acknowledge and accept their emotional experience; it targets the adaptive aversion of negative emotional experience, helping to shift attention away from emotionally salient cues or stimuli that exert negative affect by engaging in adaptive self-soothing or implementing more adaptive ER strategies. The negative affect associated with emotionally salient cues or stimuli can be changed via new learning experiences, usually in the practical form of exposure [65].

Evidentially, DBT has been found to reduce experiential and neurological emotional reactivity, as demonstrated by reduced activity in brain regions associated with emotional sensitivity and response, including the amygdala, in people with BPD, when presented with emotional stimuli [64]. Furthermore, it enhances emotional modulation, as evidenced by increased limbic–frontal connectivity and increased grey matter volume in brain regions critically implicated in ER in people with BDP, following DBT. Yet, the mechanisms of change underlying DBT (including improvements in ER) remain, for the most part, empirically unspecified [67]. As for CBT, DBT was also found by Sloan and colleagues [62] to significantly reduce ED (d = 1.28, larger effect compared to CBT; with medium effects detected at 3-month follow-up). It also had positive effects on depression, substance use, and eating disorder symptoms, and only small effects (d = 0.32) on BPD symptoms pre-post treatment, compared to both waiting list and activity-based support groups.

The importance of focusing on factors underlying ED is evidenced by the fact that DBT programs for parents have been developed and trialed to prevent and treat ED in children and adolescents [64]. Reduced parental ED and improvements in ER, in fact, can facilitate the reduction or the eradication of maltreatment perpetration, improved parenting skills, and enhanced ER modelling, protecting the child against the development of maintenance of dysregulated behavior and helping children to implement more adaptive ER skills and strategies. Although only limited research has focused on this, at least until now, it has been found that parental DBT training increases positive parenting behaviors, especially when it includes psycho–education components associated with ER strategies and skills [64].

#### 5.1.3. Schema Therapy and Acceptance and Commitment Therapy

Schema therapy has been investigated as potentially beneficial to reduce ED since it has been previously found an effective intervention for personality disorders [68]. Schema therapy is an integrative psychotherapy combining techniques and theories from other programs, including CBT and attachment theory. A *schema* can be conceptualized as a negative or biased belief about life (usually arising from early negative experiences that become *reactivated* later in life), which results in fixed and maladaptive patterns of thoughts, feelings, and coping behaviors. The main aim of schema therapy is substituting negative schema with more adaptive ones. Elements commonly targeted in schema therapy parallel Gross’ classification of the different steps of ER [2], which are situation selection, situation modification, attentional deployment, appraisal of stimuli, and response modulation. Although one important element of schema therapy is the process of “re-parenting”, whereby the patient learns and internalizes ER strategies as modelled by the therapist, the actual mechanisms by which schema therapy improves ED remain largely unknown [68].

Acceptance and commitment therapy (ACT) is a modern therapeutic approach that specifically targets psychological inflexibility. It has been proven effective for people with disorders where inflexibility plays a crucial role, such as anxiety disorders and depression [69]. ACT, instead of merely asking the patients to change patterns of thoughts, feelings, and behaviors (as other therapies do), encourages them to implement acceptance and mindfulness strategies in parallel with commitment and behavior-change strategies, with the main goal of increasing psychological flexibility. Previous systematic reviews [62,70] found that ACT is effective for reducing ED and associated behaviors, such as dysregulated eating behaviors. Sloan and colleagues reported that, compared to TAU, ACT and TAU produced significantly greater reductions in ED, with a large effect (d = 0.98).

#### 5.1.4. Mindfulness-Based Therapy

Mindfulness can be defined very simplistically as an approach aimed at limiting any judgements of emotions and/or experiences, and distancing from such experiences by focusing and directing attention to the present moment. A systematic review consisting exclusively of RCTs, by Sancho and colleagues [71], investigated the efficacy of mindfulness-based interventions (MBIs) for treating substance and behavioral addictions, of which ED is a crucial component. They found that MBIs are effective for generally reducing perceived stress, improving planning and decision-making performance, and increasing emotional and cognitive awareness. However, these effects often did not persist beyond the intervention period. Particularly when combined with CBT, MBIs showed improvements in ER and distress tolerance, and led to reductions in depression and anxiety symptoms. Mindfulness-based relapse prevention (MBRP) was found effective in promoting specific adaptive behaviors, such as self-soothing and self-regulating behaviors, reducing physiological distress and protecting against stress. Lastly, yoga—which includes some components of mindfulness—was found to improve emotional self-control, especially in women. Major limitations of previous research conducted on mindfulness-based interventions is often the lack of rigorousness (e.g., no control arm/group), making it difficult to understand the true efficacy (i.e., the effect size) of such interventions.

#### 5.1.5. Unified Protocol for Emotional Disorders

The Unified Protocol (UP) for Emotional Disorders is a transdiagnostic intervention tackling ED and neurotic tendencies. The main goals of the UP are helping the patients to understand and tolerate their emotional experiences, understanding what the situational and personal factors associated with ED are, and learning to adopt more adaptive ER strategies by changing established maladaptive approaches [72]. Although it is primarily based on CBT, the UP incorporates other features, including a focus on understanding the antecedents and consequences of ED without changing them, therefore integrating components of mindfulness-based therapy. It has been demonstrated that UP is beneficial for promoting the implementation of adaptive ER strategies (e.g., better cognitive reappraisal and increased mindfulness and awareness of emotional states and reactions), also reducing ED and the use of maladaptive strategies (e.g., emotional avoidance and suppression) [73].

#### 5.1.6. Group-Based Interventions

Moore and colleagues [74] reviewed the literature, including several study designs, on group interventions aimed at improving ER, with clinical populations. Fifteen studies with varied sample populations were included, and were focused on group-based ACT, DBT, CBT, mindfulness training, and emotion-focused psychotherapy. Most interventions were based on proactive skills and the core components included emotional psychoeducation; teaching strategies to improve emotional awareness, monitoring, and responding to emotional stimuli; and mindfulness. The importance of practicing the skills outside the group was emphasized in most studies, e.g., allocating intervention-related homework and encouraging participants to practice specific tasks in their own time. Overall, it was found that, regardless of the type of intervention, group settings were beneficial for improving ER skills, compared to waiting list or treatment as usual, with predominantly large effect sizes. However, when ER-focused interventions were compared with non-specific or more generic interventions, there were no differences in such effects. A major limitation is the fact that, in most studies, participants were allowed to engage with other interventions besides the group intervention, making it difficult to disentangle the benefits of specific group ER-focused interventions and other activities participants were involved in.

The meta-analysis by Sloan and colleagues [62] included some studies on emotional regulation group therapy (ERGT), an ER skills-based intervention, and found that ERGT was effective for significantly decreasing difficulties in ER and improving ED (with large effects), alongside improving psychopathological symptoms, including BPD, depressive, deliberate self-harm, and anxiety symptoms. Moreover, ERGT was found to significantly reduce the implementation of avoidance mechanisms, which is considered a maladaptive ER strategy (large effect size).

#### 5.1.7. Mobile Health Interventions

Digital interventions (e.g., delivered via mobile or computer apps and software) have recently increased in popularity, due to their higher accessibility and inclusivity compared to more traditional in-person interventions. Diano and colleagues [75] carried out a systematic review on mobile health (mHealth) technology apps as an adjunct to traditional psychological treatments for ED. Nineteen studies were included, consisting of six RCTs, four usability trials, four open trials, three feasibility trials, a pre- and post-intervention study, and a longitudinal qualitative study. In the context of adjunctly targeting ED, mHealth apps primarily aim to improve the integration of clinical treatment, whereby ER skills and training delivered by the therapist can be enriched and utilized via using the supportive digital environment between sessions. Prominent therapeutic factors associated with using mHealth apps as adjunct to psychological interventions include enhancing the patient’s agency, improving emotional management, and facilitating the generalization and more frequent application of acquired skills. All mHealth apps included in the review by Diano and colleagues [75] integrated or were paired with mindfulness practices, and all the interventions delivered via the mHealth apps belonged to the CBT framework or were theoretical derivations of CBT in the form of DBT or ACT. Overall, ED improvements were slightly greater when mHealth apps were used conjunctively in parallel with traditional psychological treatments, but it was difficult to discern between the efficacy of the interventions themselves and the effects associated with delivering such interventions in a digital/mobile environment [75].

#### 5.1.8. Virtual Reality-Based Interventions

Colombo and colleagues [76] reviewed the literature on virtual reality (VR)-based interventions for improving ER. These interventions—at least in theory and only based on limited literature—are designed to facilitate the acquisition and development of ER skills in extremely realistic, personally significant, and interactive virtual settings. For example, within VR environments that can be personalized and controlled by the therapist, emotional states can be effectively induced, and patients can therefore experiment implementing specific ER strategies in a safe setting. For example, situational VR interventions, involving exposure and/or behavioral activation protocols, are feasible and allow for the planning of individualized interventions, since personally relevant and realistic environments can be materialized for the participant to practice emotional responses to emotionally eliciting stimuli. Exposure-based situational VR interventions have been found effective in supporting adaptive responses to emotional content and reducing avoidance behaviors [76].

VR interventions training attentional mechanisms, such as mindfulness and relaxation protocols, have also been developed [76]. For example, mindfulness-based VR interventions have been shown to facilitate and improve the acquisition and fostering of mindfulness skills, and to significantly reduce anxiety levels on both physiological and self-reported measures. Moreover, VR-based self-compassion interventions have been found to increase awareness and present-moment attention to mental and bodily experiences. Relaxation-based VR interventions can digitally transport an individual experiencing emotional distress to a virtual environment incorporating non-arousing, peaceful, calming audio and visual stimuli, which can be interactively experienced from a first-person perspective. These have been found to reduce distressing moods and induce desirable moods, promote the learning of relaxation techniques, and facilitate stress recovery in immersive and non-immersive environments, as robustly established using physiological and self-reported measures.

Several VR interventions aimed at promoting cognitive reappraisal, which is considered an adaptive ER strategy, have been developed [76]. For example, gamified VR interventions specifically targeting cognitive distortions and promoting the acquisition and development of other cognitive skills implicated in ER, are effective for reducing depressive symptoms, particularly when personalized. Similarly, VR interventions to improve response modulation strategies have been trialed [76]. Stress inoculation interventions involve the repeated and progressive exposure to stressful stimuli to augment emotional reaction to decrease emotional activation and therefore reduce the impact on the individual’s affect. Stress inoculation interventions, paired with VR training, have been found effective for reducing stress levels and increasing resilience to stress. Furthermore, some VR interventions target impulse control, whereby VR is utilized to expose individuals to personally high-risk stimuli for teaching and development of self-regulation and ER strategy implementation. VR exposure therapy—used adjunctively with CBT—has been found to reduce phobic emotional responses by supporting participants in identifying distorted beliefs and developing more functional/adaptive thinking patterns and/or beliefs. In brief, VR exposure therapy can act as an opportunity to train and practice the implementation of ER strategies in safe and undangerous settings.

#### 5.1.9. Heart Rate Variability Biofeedback

Heart rate variability biofeedback (HRVB) is a biofeedback intervention aimed at training cardiac activity regulation and increasing HRV via specific breathing patterns [77]. Increasing HRV is theorized to be beneficial for promoting a regulatory system that can dynamically control body functions in response to various and changing stimuli which can ordinarily be destabilizing. Thus, HRVB is likely to have pervasive beneficial effects, as a result of increasing HRV, including better ER. Practically, HRVB involves controlled breathing, whereby individuals are taught or instructed to breathe slowly or slower than usual, according to the rate of the baroreflex rhythm, via biofeedback loops [77,78]. The loop informs the individual what breath rate to follow, whereby biofeedback detects the relevant, individualized baroreflex frequency that maximizes the individual’s HRV.

Ter Hamsel and colleagues [78] presented an overview of at-home HRVB, training using portable devices connected to personal computers, consisting of experimental and pilot studies comprising various study designs, including case studies, quasi-experiments, and, mostly, RCTs. The extensive technological advancements over recent years have made the implementation of HRVB more accessible and inclusive, e.g., by using non-invasive portable devices like wrist sensors or chest bands. This would allow, for example, the implementation of bio-cueing, namely, warning users when physiological markers (including but not limited to HRV) are in specified ‘at-risk’ ranges, to encourage real-time or just-in-time behavioral adjustments such as breathing at specific rhythms, as trialed during the HRVB training, resulting in better regulation of the physiological mechanisms underlying ER and ED in everyday life.

A meta-analysis by Lehrer and colleagues [77] analyzed 58 RCTs assessing the effect of HRVB on various symptoms and domains of functioning. Overall, they found HRVB and pace breathing (PB) interventions to significantly improve ER, with a small-to-medium effect (Hedge’s g = 0.34). Furthermore, the review by ter Harmsel and colleagues [78] included 26 ambulatory biofeedback studies and four bio-cueing studies, and found that the use of bio-cueing in non-psychiatric pediatric populations reduced difficulties with verbalizing emotions, allowing participants to take time to adjust their ‘triggered’ and ‘automatized’ behaviors, and increased self-control and anger management. Bio-cueing was also found effective for reducing PTSD symptomatology amongst war veterans with a formal diagnosis of PTSD when administered alongside regular anger management therapy, but it did not outperform TAU.

Most studies included in the review by ter Harmsel and colleagues [78] used ambulatory HRVB in non-psychiatric populations. It was found that this type of intervention produced significant increases in HRV and reductions in HR. There were also significantly greater reductions in psychological stress measures, distress, negative affect, and depression, and burnout symptomatology, and significant improvements in quality of life, compared to waitlist controls. Nevertheless, ambulatory HRVB was also found as effective as mindfulness meditation and physical exercise in improving stress-related outcomes in some populations. In pregnant females, ambulatory HRVB was found to significantly reduce self-reported childbirth-related anxiety, predicting lower pre-natal and post-natal depression scores. Amongst psychiatric populations, it was found that ambulatory HRVB is effective in reducing substance cravings and depression in young adults with substance use disorder, but at similar levels to music relaxation. Moreover, significant reductions in panic severity, global illness severity, and functional impairment following a brief ambulatory HRVB intervention were found in individuals with panic disorder. Similarly, this intervention was effective in reducing state anxiety in those with disorders associated with sympathetic over-arousal, which notably includes generalized anxiety disorder and obsessive-compulsive disorder.

#### 5.1.10. Food Supplements

We identified only a meta-analysis of RCTs of food supplements and their effects on ED. This work by Karaszewska and colleagues [79] consisted of four RCTs and showed benefits associated with marine mega-3 fatty acid (i.e., fish oil) supplementation on affect dysregulation in adults with BDP (with a large effect size). Considering that deficiencies in omega-3 fatty acids have been associated with violent behavior and emotional lability (related to consequent alterations in serotonin and dopamine levels) [80], omega-3 supplementation might eventually prove helpful for reducing ED, but evidence is scarce and further research is needed before making solid conclusions.

### 5.2. Children and Adolescents

Amongst the management and treatment options for ED in children and adolescents, psych–social interventions, notably parent management training (PMT) and psychoeducational interventions for young people and their families, are common [15]. Founded on the principles of social learning theory, PMT teaches caregivers techniques and strategies based on behavioral reinforcement, therefore aiming to reduce problematic behaviors (by implementing non-reinforcing consequences) and increase prosocial behaviors (via positive reinforcement techniques). Improvements in ED have been found as secondary outcomes of psycho–social interventions that conventionally target disorders without an explicit focus on ED [15]. Notably, parent–child interaction therapy facilitates significant improvements in emotional functioning, by teaching the caregiver to attend to and address their own emotional reactions to their child’s behavior first. This allows the caregiver to act as a coach, guiding their child toward emotional recognition and supporting the development of ER skills.

Other psychological interventions have been found to reduce ED and improve ER in children and adolescents. Moltrecht and colleagues [81] meta-analyzed the results of 21 RCTs assessing changes in ED and ER following psychological interventions (e.g., CBT or CBT-based interventions, motivational interviewing, ER skills training, ACT, and problem-solving-based programs, compared to treatment as usual or waitlist controls) in children and young people aged 6–24 years old. They found that these interventions were beneficial in reducing ED (moderate effect) and improving ER (small effect). Importantly, it was found that ER-focused interventions were slightly more effective than non-specific interventions (e.g., CBT) in reducing ED, but non-specific interventions were slightly more effective than ER-focused interventions in improving ER. The effects of these interventions (both for reducing ED and improving ER) were also reduced in studies with active control conditions compared to passive. Lastly, intervention-related reductions in ED partially explained reductions in deliberate self-harm and general improvements in psychopathology.

A larger meta-analysis by Daros and colleagues [82] studied the effectiveness of psychological interventions (both RCTs and non-RCTs) on ER and ED, in young people with depression and anxiety. Ninety studies were included (11,652 participants in total, aged 14–24), trialing a wide range of interventions including CBT- or CBT-based, mindfulness-based, cognitive training, acceptance/ER-based (i.e., DBT and ACT), psychodynamic, and family interventions. As in the study by Moltrecht and colleagues [81], small-to-medium effects were found in relation to decreases in ED and improvements of ER skills. This study also highlighted a statistically significant association between decreased ED, better ER skills, and improved symptoms of depression and anxiety, in line with the study by Moltrecht and colleagues [81] and the idea that improvements in ER (especially when ER and ED are targeted by the intervention itself) underlie transdiagnostic improvements in psychopathology and mental wellbeing.

Studies have also been conducted in children and adolescents in residential and acute inpatient settings. These young people often manifest severe psychopathology and dysregulation; therefore, they may benefit from settings and interventions that provide safety and stabilization [15,83]. In such settings, it might be challenging to accurately determine the efficacy of programs for treating ED and improving ER. Nonetheless, it has been shown that behavior-management programs utilizing a social-learning approach (with clear expectations for the child and positively reinforcing positive behaviors) reduce ED [15,83]. Similarly, utilizing a rewards system and individualized behavior plans, modified positive behavioral interventions and supports have been found to reduce the frequency and duration of ED displays [15]. Likewise, collaborative problem solving (CPS) as a caregiver-mediated behavior modification strategy has been shown to reduce the frequency and duration of ED displays [15,83].

Reynard and colleagues [84] conducted a meta-analysis investigating the efficacy of digitally based interventions to promote ER in children and adolescents. This is a particularly important area of research, since more traditional psychological interventions for ER and ED are oftentimes inaccessible to some children and young people. The review consisted of 11 RCTs (2476 participants) focusing on biofeedback, digital games, VR and augmented reality, and program and multimedia interventions. The study found that digitally based interventions (when taken together) did not produce any statistically significant changes in ER or ED. Nevertheless, digital game interventions were found to significantly improve emotional experiences (with a small effect) but did not have any effects on ER skills.

In adolescents with anger management problems and disruptive behaviors, bio-cueing was found effective for reducing anger and aggressive behaviors, and this type of intervention was well received by participants and their caregivers [78].

Lastly, Cooper and colleagues [80] conducted a meta-analysis on 10 studies, including only randomized double-blind placebo-controlled trials, investigating the effect of omega-3 supplementation on emotional and mood lability, irritability, oppositional behaviors, and conduct problems, in children with a diagnosis of ADHD and related neurodevelopmental disorders. They found that omega-3 supplementation was not more effective than placebo in improving emotional lability, as rated by parents or teachers.

## 6. Clinical Implications and Future Directions

We narratively reviewed the evidence about the effectiveness of pharmacological and non-pharmacological interventions for reducing emotion dysregulation (ED) and improving emotion regulation (ER). Overall, based on the current evidence, the most used and effective are medications commonly used to treat depression (both SSRIs and NRIs), ADHD (both stimulants and non-stimulants), and psychotic symptoms (e.g., risperidone or aripiprazole); and psychological interventions or adaptations of such interventions (based on CBT, DBT, and mindfulness components), e.g., by integrating biofeedback components (HRV biofeedback) or delivered via digital technologies (e.g., mobile or virtual reality).

From a clinical standpoint, the evidence is insufficient to provide clear guidelines for clinicians and mental health practitioners. Studies assessing the specific effects of each intervention on the bio–psycho–social factors that have been proposed to contribute to the onset and maintenance of ED and difficulties in ER are needed to inform clinical practice. Unfortunately, our review did not identify any studies—specifically focused on ED or ER—addressing this. However, there is preliminary evidence that the function and activation of frontal–limbic systems involved in emotion processing and/or regulation is changed with CBT and antidepressants in people with depression [85,86], with CBT and SSRIs in people with anxiety [87], with stimulants and non-stimulants in people with ADHD [88], with DBT in people with BPD [89], and with mindfulness-based interventions [90,91]. In relation to HRV, it has been shown that HRV biofeedback produces increases in HRV, and this is reflected in reduced ED and better ER at the behavioral level [77,78]. A major limitation of the current literature is that methodological differences across studies (e.g., type of intervention and duration, active vs. passive control arm, sample characteristics, and type of imaging technique used) often lead to inconsistent patterns and findings that are rarely replicated in larger follow-up studies. Adopting transdiagnostic and computational approaches to the study of the effects of different interventions for ED on brain and body functioning could help predict what intervention works best for people with specific clinical and demographic profiles.

We have identified some limitations of both our work and the reviewed literature, which shall be addressed in future research. This is a narrative review, and our search (although systematic) was restricted to systematic reviews and meta-analyses, which could have led to selection bias, namely, the selection of some studies and the possible exclusion of others. While systematic reviews and umbrella reviews studying interventions for ED have been published (see, for example, [60]), these are often limited to one interventional domain (e.g., pharmacological or non-pharmacological), one type of intervention, and/or specific disorders (which collides with the theories about the transdiagnostic nature of ED). Further evidence synthesis studies, e.g., individual participant data network meta-analyses, are needed to elucidate the effects of different pharmacological and non-pharmacological interventions both on ED symptoms at behavioral level and in relation to underlying bio–psycho–social factors. Lastly, most of the systematic reviews and/or meta-analyses included cross-sectional studies, which cannot provide conclusive information about the maturation of ER skills across development and the factors affecting such developmental trajectories. For example, the fact that ER circuits continue to mature until young adulthood highlights the importance of planning future studies aimed at understanding developmental trajectories of ER skills maturation, and explaining why differences in emotional processing and regulation between younger adolescents and older adults are often evident.

## 7. Conclusions

Overall, we found evidence of beneficial effects on both reducing ED and improving ER skills for antidepressants, ADHD medication (both stimulants and non-stimulants), and antipsychotics, but also for cognitive and dialectical behavioral therapy, schema therapy, acceptance and commitment therapy, mindfulness-based interventions (especially when combined with psychotherapy), heart rate variability biofeedback, and psycho–social interventions involving caregivers and whole family systems. We also found that implementing non-pharmacological interventions in digital environments (e.g., via mobile apps or virtual reality) is likely to be as effective as in-person interventions.

We think that advancing our knowledge about specific factors underlying ED could support the identification of novel or more specific interventions for ED. For example, identifying specific neurobiological and neurophysiological markers of irritability could help pave the way for innovative pharmacological treatments for ED. Furthermore, psycho–social factors and the impact of negative early life experiences are more likely to be targeted by non-pharmacological approaches (especially, psycho-therapeutic and psycho-social interventions). Further research shall therefore investigate the specific effects of pharmacological, non-pharmacological, and combined interventions in transdiagnostic samples of children, young people, and adults with different diagnoses. In parallel, within the area of precision medicine, it would also be important to investigate if specific interventions work better for certain subgroups (e.g., those presenting with internalized vs. externalized ED), with the ultimate goal of identifying effective strategies to nurture ER and reduce ED.

## Data Availability

Not applicable.

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
